# Comparison between Image-Guided Transbronchial Cryobiopsies and Thoracoscopic Lung Biopsies in Canine Cadaver: A Pilot Study

**DOI:** 10.3390/ani12111388

**Published:** 2022-05-28

**Authors:** Ilaria Falerno, Roberto Tamburro, Francesco Collivignarelli, Leonardo Della Salda, Luigi Navas, Rossella Terragni, Paolo Emidio Crisi, Andrea Paolini, Francesco Simeoni, Massimo Vignoli

**Affiliations:** 1Faculty of Veterinary Medicine, University of Teramo, 64100 Teramo, Italy; rtamburro@unite.it (R.T.); fcollivignarelli@unite.it (F.C.); pecrisi@unite.it (P.E.C.); apaolini@unite.it (A.P.); fsimeoni@unite.it (F.S.); mvignoli@unite.it (M.V.); 2Department of Veterinary Medicine and Animal Production, University of Naples Federico II, 80137 Naples, Italy; luigi.navas@unina.it; 3Clinica Veterinaria Pet Care, 40133 Bologna, Italy; terragni.rossella@gmail.com

**Keywords:** image-guided lung biopsy, transbronchial lung cryobiopsy, CT fluoroscopy, interstitial lung disease, small animals

## Abstract

**Simple Summary:**

A definitive diagnosis for most pulmonary diseases is possible only through histopathological examination. The literature describes different methods of lung biopsy sampling depending on the case. However, for the diagnosis of diffuse interstitial pulmonary diseases and some peripheral neoplasms, the gold standard is represented by surgical lung biopsies. Given their invasiveness and the high percentage of risk for the patient, in most cases they are not carried out, resulting in a serious diagnostic gap. In human medicine, transbronchial lung cryobiopsies have been introduced as an alternative, which have shown high efficacy and reduced invasiveness. This study aims to evaluate the feasibility of the new technique in dogs by subjecting dog cadavers to pulmonary cryobiopsy and surgical lung biopsies, and to compare the samples obtained for histopathological quality. In total, 42 tissue samples were compared. Pulmonary cryobiopsies were smaller than surgical biopsies but with high levels of agreement upon histological evaluation. This study demonstrates the feasibility of the technique in dogs and the collection of specimens with size and histological features comparable to those from surgical biopsies.

**Abstract:**

To date, the only method of sampling lung tissue with a high diagnostic yield is represented by surgical lung biopsies (SLB), which are highly invasive and have a high risk/benefit ratio. In humans, transbronchial lung cryobiopsies (TBLC) have recently been introduced, which are described to be less invasive and able to significantly increase diagnostic confidence in most patients with interstitial lung diseases. The aim of this study was to evaluate the feasibility and diagnostic yield of TBLC compared to SLB in small animals. A total of 21 pulmonary cryobiopsies under fluoroscopic and real-time CT fluoroscopic guidance and 21 video-assisted thoracoscopic surgery (VATS) lung biopsies were collected from three dog cadavers. Upon histological examination, cryobiopsy samples were smaller than VATS biopsies, but were still large enough to reach a specific diagnosis or to allow pattern recognition. Morphological features on TBLC and SLB were concordant in all cases. Cryobiopsy samples showed fewer artifacts and a higher percentage of alveolar tissue than VATS samples. TBLC is a feasible and useful alternative to SLB for lung histopathological examination in dogs. The effectiveness and reduced invasiveness of TBLC compared to SLB could represent many advantages in the diagnosis of diffuse lung diseases in small animals.

## 1. Introduction

Diagnosing pulmonary diseases in small animals can often be a challenge. The diagnostic tools available do not always allow for the achievement of a definitive diagnosis, leading to therapeutic failures and poor prognosis. A histopathological diagnosis is often necessary, in particular for diffuse interstitial lung diseases such as idiopathic pulmonary fibrosis (IPF) or IPF-like conditions, pulmonary neoplasia, and in all cases where other diagnostic options have failed [1,2,3,4,5].

Interstitial lung diseases (ILD) are a heterogeneous group of non-neoplastic and non-infectious diseases characterized by inflammation and fibrosis of the lung parenchyma. The recently proposed classification scheme of ILD in small animals, translated from human medicine, includes three groups: idiopathic interstitial pneumonias (IIP), ILD of known causes, and miscellaneous ILD [2]. Idiopathic pulmonary fibrosis is the most described IIP disease in small animals. However, the studies in the literature were conducted on a limited number of cases, with diagnoses mostly presumed or obtained at necropsy [6]. IPF is a chronic, progressive, ominous disease of unknown etiology. History, clinical findings, laboratory tests, and diagnostic imaging can lead to a suspected diagnosis. However, in animals, the definitive diagnosis still requires a histological confirmation, often obtained post-mortem [2,3,6]. Clinical examination findings include cough, exercise intolerance, respiratory distress, inspiratory crackles, cyanosis, and syncope. Results of serum biochemistry, complete blood count, endoscopy, and bronchoalveolar lavage are not specific for ILD but allow for other diseases to be ruled out. Similarly, thorax radiographs are non-specific in patients with ILD. The diffuse bronchointerstitial pattern is more often represented, but bronchial and alveolar patterns have also been reported. Doppler echocardiography is recommended in the evaluation of pulmonary hypertension (PH), a frequent comorbidity in these patients [2,6,7,8,9,10,11]. Useful information can also be provided by thoracic computed tomography (CT) single or multi-phase angiography, which can identify pulmonary vascular diseases, pulmonary parenchymal diseases, or pulmonary thromboembolism as underlying causes of PH, as well as findings indicative of PH. CT pulmonary trunk-to-descending aorta ratio ≥ 1.4 suggests moderate and severe pulmonary PH in dogs [12,13].

High-resolution computed tomography (HRCT) is certainly the most sensitive imaging modality in evaluating the lung parenchyma. In humans, years of multidisciplinary collaboration between clinicians, diagnosticians, and pathologists have allowed for the characterization of lung patterns in HRCT. Usual interstitial pneumonia (UIP) pattern detection in HRCT is diagnostic of human IPF. A honeycomb pattern with or without bronchiectasis or peripheral traction bronchiolectasis is the distinctive feature of UIP, with typical subpleural distribution. In this case, lung biopsy is no longer necessary [14].

In canine IPF, the most common CT findings are ground-glass opacity, mosaic attenuation, and reticular patterns, while traction bronchiectasis and honeycombing are rarely described and are associated with later stages of disease [7,8,15,16,17]. Due to the paucity of correlation data between CT and histopathological patterns for ILD in the veterinary literature, histopathology of peripheral lung is the gold standard for diagnosis, as imaging results in the absence of biopsy confirmation might be misleading [15,18].

Lung biopsies can be performed by bronchoscope biopsy, needle biopsy, and surgical lung biopsy (thoracoscopic surgery or open thoracotomy) [19]. The choice of method is made according to the location and characteristics of the biopsy samples that are needed [4,20,21,22]. To date, for the histological diagnosis of diffuse interstitial lung diseases, the only method of sampling peripheral lung tissue with a high diagnostic yield is represented by surgical lung biopsies (SLB), which are highly invasive and have an unfavorable risk/benefit ratio [2]. SLB can be done by video-assisted thoracoscopic surgery (VATS) or open thoracotomy. VATS lung biopsy is minimally invasive surgery that allows for the collection of peripheral lung specimens with a recommended distance of less than 2–3 cm from the edge of the lung lobe [21]. It is usually performed by multi-portal technique using an intercostal or transdiaphragmatic approach in patients undergoing general anesthesia and one-lung ventilation. Loop ligatures, biopsy forceps, endostapler devices, or vessel-sealing devices can be used [4,21,22]. In humans, VATS is preferred over open thoracotomy in patients requiring SLB, if they can tolerate one-lung ventilation [14].

In human medicine, transbronchial lung cryobiopsies (TBLC) have recently been introduced [23], which are described to be less invasive than SLB, with manageable side effects and the ability to significantly increase diagnostic confidence in most patients with ILD [24,25]. TBLC are performed by bronchoscope under fluoroscopic guidance to target the subpleural area of interest. The absence of fluoroscopy guidance during the procedure was related higher complication rates [26]. As an alternative to conventional fluoroscopy, real-time CT fluoroscopy has been proposed as a guide for TBLC [27]. Real-time CT fluoroscopy is described in humans as an alternative to traditional CT or conventional fluoroscopy for interventional procedures such as lung biopsy and fluid drainage, allowing us to obtain CT scans in real time and to view them directly on a monitor in-room during the CT-guided procedure [27,28].

The aim of this pilot study was to evaluate the feasibility and diagnostic yield of TBLC compared to VATS lung biopsy in dogs. This is the first step of a study on transbronchial lung cryobiopsies in small animals.

## 2. Materials and Methods

This study was approved by the Ethical Animal Care and Use Committee of the University of Naples Federico II (n. PG/2020/0079565).

Two fresh medium-sized canine cadavers and one medium-sized dog that had died 24 h earlier underwent a lung biopsy procedure using both imaging-guided TBLC and VATS. The dogs had been euthanized for causes unrelated to lung disease. The three dogs were placed and kept in sternal recumbency for all the procedures they underwent. The fresh cadavers immediately after euthanasia were intubated with an endotracheal tube connected to an anesthetic machine with a mechanical ventilator. The ventilator was set to keep the lungs inflated at a pressure of 12 cm H_2_O. Ventilation was interrupted during TBLC and VATS lung biopsies.

Before the biopsies, a CT study of the thorax was performed in the two fresh cadavers. CT images were viewed at a window setting optimized for lung parenchyma (width, 1500 HU; level, −500 HU), with soft tissue and lung algorithms, and a slice thickness of 1.3 mm. An HRCT scan was performed with an axial scan, 0.6–1.3 mm slice thickness, and 300–350 mAs. A CT-fluoroscopy was performed with parameter settings of 40 mAs, 120 Kv, and 2.5 mm slice thickness.

TBLC were performed first. A flexible cryoprobe measuring 115 cm in length and 1.7 mm in diameter was used (ERBE, Tübingen, Germany). The probe was cooled with carbon dioxide. The cadavers were extubated. The cryoprobe was inserted via flexible bronchoscope with a working channel inner diameter of 2.3 mm and a working length of 85 cm (Storz, Guyancourt, France) under real-time CT fluoroscopic guidance (Optima 540 GE, Milwaukee, WI, USA) for the two fresh cadavers and conventional fluoroscopic guidance (high-frequency beta tube generator, table delta 90, CAT Medical Systems, Rome, Italy) for one cadaver.

Real-time CT fluoroscopy allows for the reconstruction and visualization of the acquired CT data in real time. The fluoroscopy and CT fluoroscopy images were displayed directly on an in-room monitor. Sampling sites were randomly chosen from the left and right caudal lobes, the right middle lobe, and the left and right cranial lobes while targeting the peripheral lung in the subpleural area.

Once the sampling site was reached, the cryoprobe was cooled by pressing the foot pedal of the cryo unit (ERBECRYO2, ERBE, Tübingen, Germany) for a mean time of 8 s, and the frozen sample was extracted while attached to the probe’s tip. 15 CT fluoroscopy-guided TBLC (10 and 5 samples from the two fresh cadavers, respectively) and 6 fluoroscopy-guided TBLC were collected. The frozen sample was thawed via saline irrigation and transferred to formalin for fixation. At the end of the TBLC procedure, the cadavers were reintubated and kept under mechanical ventilation. A post-procedure thorax CT study was performed in the two fresh cadavers.

Then, in the same room, a standard 3-portal VATS lung biopsy was performed according to the technique described in the literature [22]. An endoscopic cutting device—the EnSeal (Ethicon Endo-Surgery, Blue Ash, OH, USA)—was used. Peripheral lung samples were randomly taken from the left and right caudal lobes, the right middle lobe, and the left and right cranial lobes. 10 and 5 samples from two fresh cadavers, respectively, and 6 samples from one cadaver were collected. The tissue specimens were fixed in formalin for histological examination.

A first macroscopic evaluation of the size of all samples was performed before being transferred to formalin.

The lung biopsies were promptly fixed in 10% neutral-buffered formalin for histopathological investigations. Paraffin-embedded 4 µm-thick sections were routinely processed for histology and stained with hematoxylin–eosin (H.E). For the acquisition of the histological serial images, a Leica DMRE light microscope was used, and the evaluation of the length and area of the slides was carried out using the image analysis software LAS X Measurements (Leica microsystems, Heerbrugg, Switzerland).

For an analysis of the TBLC compared to VATS lung biopsies data, the surface area of tissue samples, the presence of pleural tissue and alveolar tissue, and the presence of artifacts related to the procedure (“crush” artifacts) were recorded.

Data of the morphometric area were assessed for normality using a D’Agostino and Pearson test; data were not normally distributed, and a Mann–Whitney Test was used with a *p*-value set at <0.05.

## 3. Results

TBLC and VATS lung biopsies obtained from the two fresh dog cadavers (weight: 25 kg and 20 kg) and the one dog who had died 24 h earlier (weight: 22 kg) were evaluated. In total, 21 TBLC and 21 VATS lung biopsies were collected and compared.

TBLC were performed via flexible bronchoscopy under image guidance from different areas of lung lobes. With the advancement of the flexible cryoprobe through the working channel of the flexible bronchoscope, peripheral lung samples in the subpleural location were extracted. A CT study of the thorax of the two fresh canine cadavers before biopsies showed no lung collapse and no pathological changes. TBLC under real-time CT fluoroscopic guidance was performed (Figure 1). A real-time CT fluoroscopy was used for direct cryoprobe visualization during advancement and sampling in the subpleural area. Fifteen lung cryobiopsies were obtained, comprising 10 and 5 samples from the two fresh cadavers, respectively.

The dog who had died 24 h earlier was subjected to TBLC under conventional fluoroscopic guidance; fluoroscopy allowed the direction of the cryoprobe towards the subpleural site, displaying monoplane X-ray images in real time. Six pulmonary cryobiopsies were collected.

In both procedures, three operators participated, two of whom were within the X-ray room and equipped with radiation protection devices. Double-sided lead aprons, thyroid collars, goggles, and lead surgical gloves were used. During CT fluoroscopy, the CT scanner provides a real-time radiation dose report by recording the CT dose index (CTDI) and dose-length product (DLP) values for each acquired series and provides the DLP for the entire exam. No data on operator radiation exposure were recorded in this study.

After TBLC, the cadavers were subjected to VATS lung biopsies using the standard three-port technique and an endoscopic cutting device. Peripheral lung samples were collected in equal numbers to the TBLC for each cadaver, with 10 and 5 samples for the fresh cadavers, respectively, and 6 samples for the cadaver of the dog who died the day before. All 42 tissue samples were morphometrically evaluated, and 30 samples were also morpho-histologically compared (Table 1).

For the first bidimensional macroscopic evaluation (Figure 2), the mean area of the cryobiopsy samples was 28 mm^2^ (range 12–60 mm^2^). The mean area of the surgical biopsies was 48 mm^2^ (range 12–108 mm^2^) (Table 2).

A morphometric analysis of the histologic slide (Figure 3) of samples showed a mean area of 20.20 mm^2^ by TBLC compared to 46.20 mm^2^ obtained by SLB.

A significant difference was observed between the VATS lung biopsy (median 10.7 mm^2^; minimum 1.98 mm^2^ maximum 176.64 mm^2^) and the TBLC (median 21.3 mm^2^; minimum 7.41 mm^2^ maximum 408.09 mm^2^) morphometric areas (*p* = 0.0085).

Lung biopsies were then compared morpho-histologically. A total of 12 lung biopsies (6 TBLC and 6 SLB), obtained from the cadaver of the dog who died 24 h earlier, were excluded for excessive post-mortem tissue deterioration or presence in different samples of pleura and thoracic walls (VATS biopsies mainly). Morphological features on TBLC and SLB were concordant in all cases (Figure 4). Of 30 total lung samples (15 TBLC and 15 SLB) evaluated, 6.6% of TBLC (1/15) and 40% of VATS (6/15) were non-diagnostic/inadequate. The main artifacts and non-pulmonary tissues found in VATS biopsies were focal parenchymal collapse and the presence of parietal pleura, while a cryobiopsy consisted mainly of the bronchial wall and large vessels (Figure 5).

Upon histological examination, cryobiopsies were smaller than VATS biopsies but large enough to allow for pattern recognition. Cryobiopsy samples showed a lower degree of crush artifacts and a higher percentage of alveolar tissue than VATS samples.

## 4. Discussion

The diagnosis of certain pulmonary diseases is achieved only via histological examination. This is particularly true for diffuse interstitial lung diseases, for neoplastic diseases, or in cases where other less invasive diagnostic tools have failed [1,2,18]. Imaging plays a fundamental role in detecting pulmonary lesions and as a guide for performing lung biopsies [29,30].

Lung sampling techniques include, in order of invasiveness, needle biopsy, bronchoscope biopsy (endobronchial or transbronchial), thoracoscopic biopsy, and open lung biopsy [20]. SLB represents the gold-standard lung sampling method for diagnosing diffuse lung disease and some peripheral neoplastic lesions. Due to their invasiveness and their associated high risk, the possibility of obtaining a definitive diagnosis is often turned down [2,6,18].

This study describes a novel technique in veterinary medicine of obtaining a transbronchial lung biopsy using a flexible cryoprobe in dogs.

The TBLC procedure has been described in human medicine [23], and numerous studies have recognized its efficacy in the diagnostic field as a valid, less invasive alternative to surgical biopsies [31,32]. The major field of application of TBLC is in the diagnosis of interstitial lung diseases [33], for which until now the gold standard was represented by surgical biopsies [14].

In our study, the technique proved to be feasible. TBLC were obtained from the periphery of the lung lobes up to the subpleural area by directing the cryoprobe’s tip to the target points chosen in real time under conventional fluoroscopy and real-time CT fluoroscopy guidance.

Real-time CT fluoroscopy combines the advantages of traditional fluoroscopy and CT techniques, allowing the operator to view cross-sectional CT images in real time during a CT-guided procedure, increasing the accuracy of the procedure and reducing operating times, and subsequently reducing the duration of anesthesia. For lung biopsies, a 27% reduction in time is described using interrupted real-time CT fluoroscopy compared to traditional CT [34]. Furthermore, CT-fluoroscopy guidance has been described to improve the sampling targeting ability and safety profile of cryobiopsy by accurately establishing the probe–pleural relationship [27]. Procedural times were not recorded in our study due to the large and variable number of biopsy samples collected for each cadaver. However, viewing the cross-sectional CT fluoroscopy images increased the accuracy of directing the cryoprobe to the subpleural area while maintaining an adequate safe distance from the pleura, ensuring a reduced risk of post-procedural complication.

The disadvantage of CT fluoroscopy is the radiation exposure of the operators in the room [35]. Radiation protection devices (double-sided lead aprons, thyroid collars, goggles, and lead gloves) and dosimeters—particularly for the hands—should be worn during procedures [36]. However, by using a radioprotection device and low-milliampere and kilovoltage techniques, the radiation exposure can be reduced [37,38]. Several comparative studies between CT fluoroscopic guidance and conventional CT guidance for performing percutaneous interventional procedures have reported a reduced or similar patient radiation dose and reduced procedural times in CT fluoroscopy compared to conventional CT [35,39].

TBLC in humans is considered to be a less invasive and safer method than SLB. The main side effects described for both procedures are infection, bleeding, pneumothorax, exacerbation of interstitial pulmonary fibrosis, and perioperative mortality [24]. Postoperative pain is typical of SLB, to a lesser extent in thoracoscopic procedures than in thoracotomy [40]. A meta-analysis study reported incidences of 4.9% (2.2–10.7%) of moderate endobronchial bleeding and 9.5% (5.9–14.9%) of pneumothorax after TBLC, an 0.7% incidence of 30–60 days mortality post TBLC versus 1.8% with VATS, and a median hospitalization time of 2.6 days versus 6.1 days with VATS [31]. The data reported in the literature on the incidence of bleeding and pneumothorax are highly variable; however, these are manageable conditions with the endobronchial balloon and tube chest drainage approaches, respectively, and the mortality rate remains constantly lower than that of SLB [24,31,41,42]. The variability of the complication rates related to the procedure reported in the literature Is influenced by the lack of standardization of some aspects of the method. In fact, patients undergoing TBLC in numerous studies were deeply sedated, maintained in spontaneous ventilation, and intubated with a rigid tracheoscope. However, some studies report the description of the procedure performed in conscious sedation without intubation, while other studies describe the use of jet ventilation [26,33]. A higher percentage of pneumothorax was observed in intubated patients undergoing deep sedation with jet ventilation compared to patients undergoing sedation and spontaneous ventilation [43].

Before subjecting patients to TBLC, preliminary tests, including a complete blood test and coagulation profile, are performed. The presence of uncorrected coagulopathies represents one of the exclusion criteria in some studies reported in the literature [44,45]. In veterinary medicine, there is no supporting literature; however, it is prudent to perform blood tests and a coagulation profile as a screening before performing invasive procedures which include the risk of bleeding. In our study, no data relating to complications related to the procedures were recorded as they were performed on the cadavers. The degree of procedural complications needs to be assessed with in vivo studies and on a larger number of cases.

In this study, lung cryobiopsies and VATS lung biopsies were compared based on the size and histological features. Cryobiopsies were smaller than VATS biopsies (main 20.20 mm^2^ vs. 46.20 mm^2^; *p* = 0.0085) but were still large enough and of good enough diagnostic quality (93.4% of TBLC samples) to reach a specific diagnosis or to allow pattern recognition. A total of 12 lung biopsies (6 TBLC and 6 VATS) of the dog who had died 24 h earlier were excluded from morpho-histological evaluation due to poor diagnostic quality related to post-mortem tissue deterioration or presence in different samples of pleura and thoracic walls (VATS biopsies mainly). Morphological features of TBLC were concordant to SLB in all cases. Histological sections of both showed suitable alveolar spaces, i.e., not collapsed or emphysematous, a low percentage of atelectasis induced by the method or by cadaveric animals, preserved interstitium and the absence of artificial extravasation of blood. Cryobiopsies were characterized by a low grade of crush artifacts and a high percentage of alveolar tissue. Only one cryobiopsy was found non-diagnostic due to the presence of large vessels and many bronchi in a large percentage of the sample area.

The results obtained in our study are in agreement with the data reported in the human literature. In humans, cryobiopsy sizes ranging from 6.6 to 64.2 mm^2^ are reported, mainly depending on the experience of the operators and the diameter of the cryoprobe used (1.9, 2.4 or 2.7) [46,47]. In our study the TBLC main area was 20.20 mm^2^. The area of tissue samples has been shown to strongly influence the diagnostic field [45]. For an accurate diagnostic evaluation, a minimum size of biopsy samples of 5 mm was suggested by pathologists [41]. High levels of agreement between TBLC and SLB have been described [48].

In our study, there were more non-diagnostic VATS biopsies than TBLC due to the presence of pleura and thoracic walls. This is in disagreement with the data reported in the literature describing a high diagnostic yield of VATS lung biopsies. This may be explained by the presence of increased post-mortem tissue deterioration since VATS were performed after the TBLC procedure in cadaveric subjects.

In humans, TBLC were also compared to transbronchial biopsies with forceps. Forceps biopsies are smaller and characterized by a higher percentage of crush artifacts [23,44,49]. Babiak at al., 2009 reported a median area of the specimen taken with forceps on the histological slide of 5.82 mm^2^, compared to 15.11 mm^2^ obtained using a cryoprobe [23]. Transbronchial forceps biopsies were considered inadequate and had a lower diagnostic field than TBLC in diagnosing diffuse lung diseases [23,44,49]. In veterinary medicine, a sample size from 1 to 2.5 mm^2^ and the inadequacy of the technique for the diagnosis of diffuse pulmonary diseases are described [20].

This study describes the applicability of the novel image-guided transbronchial lung cryobiopsy technique in dogs, which allowed for the collection of representative lung tissue samples consistent with SLB. The size and histological features of the collected samples were consistent with the data reported in the human literature. Our study was conducted on medium-sized dogs using a flexible cryoprobe with a diameter of 1.7 mm. This cryoprobe caliber would be inadequate in small dogs or cats. However, the literature reports the introduction of a mini cryoprobe with a diameter of 1.1 mm [49], which could make it possible to carry out the method in small-sized subjects.

To our knowledge, there are no studies in the veterinary literature on real-time CT fluoroscopy and TBLC in small animals.

The limitations of the study were related to examining cadavers. In one case, this resulted in a lower diagnostic quality of the biopsy samples due to post-mortem tissue deterioration. Another limitation was related to the lack of data on procedural times and on the risks of complications due to the high and variable number of samples collected for each cadaver. In humans, to achieve the histological diagnosis of ILD, it is recommended to take two lung samples [42]. The diagnostic yield is increased with the collection of two lung samples taken from two different segments of the same lobe [33].

Finally, tissue samples were obtained from the same cadavers, which first underwent TBLC and then SLB. This affected the largest number of non-diagnostic SLB, disagreeing with the data reported in the literature.

However, image-guided TBLC, in addition to reduced invasiveness and potentially lower risks compared to SLB, would allow for the obtainment of a peripheral lung biopsy immediately following CT examination, in sternal decubitus, and without undergoing displacements, reducing anesthetic times in patients with compromised respiratory function.

## 5. Conclusions

Image-guided TBLC was found to be a feasible and useful alternative to SLB for lung histopathology in dogs. Imaging plays a fundamental role in reaching the target point. Real-time CT fluoroscopic guidance increases procedural accuracy and reduces risk and operating time. In humans, TBLC are described as less invasive, with manageable side effects and the ability to increase diagnostic confidence for interstitial lung disease in a multidisciplinary team discussion (MDTD) of clinicians, radiologists, and pathologists. Interstitial lung diseases in veterinary medicine are underdiagnosed and characterized by an important diagnostic gap due to the high risks associated with the procedures currently available [2,4,5]. Further studies on the risk assessment and diagnostic field of transbronchial cryobiopsies in the diagnosis of diffuse lung diseases in dogs and cats should be encouraged.

## Figures and Tables

**Figure 1 animals-12-01388-f001:**
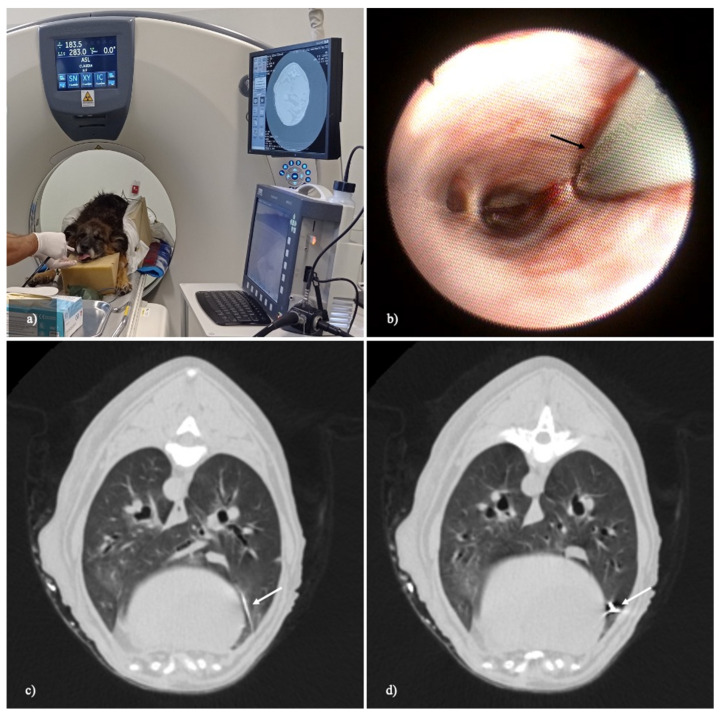
A transbronchial lung cryobiopsy via flexible bronchoscope under real-time CT-fluoroscopic guidance. (**a**) The dog is placed in sternal recumbency in the CT gantry. The flexible cryoprobe is inserted into the working channel of the bronchoscope and they are advanced together. (**b**) The flexible cryoprobe (black arrow) advances in the small bronchial branches until it reaches the target point in the peripheral lung. (**c**,**d**) Real-time CT fluoroscopy images (width, 1500 HU; level, −500 HU; slice thickness: 2.5 mm) during the flexible cryoprobe advancement. The flexible cryoprobe (white arrow) under real-time CT fluoroscopy guidance is pushed into the sampling point in the subpleural area. The acquired CT data are reconstructed and displayed in real time on the monitor in-room (**a**).

**Figure 2 animals-12-01388-f002:**
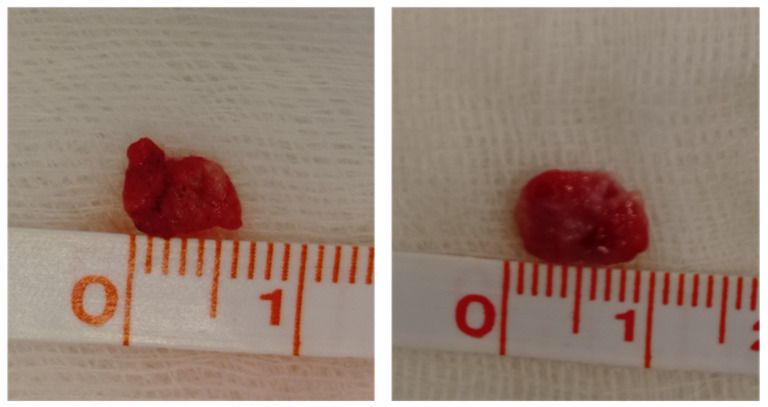
Two-dimensional measurements of two pulmonary cryobiopsies before being fixed in formalin.

**Figure 3 animals-12-01388-f003:**
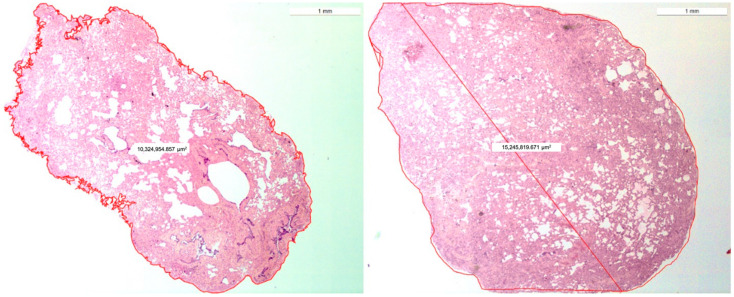
A morphometric analysis of the histological slide of a TBLC on the **left** (10.3 mm^2^) and a VATS lung biopsy on the **right** (15 mm^2^). (Ob. 2.5×) H.E stain.

**Figure 4 animals-12-01388-f004:**
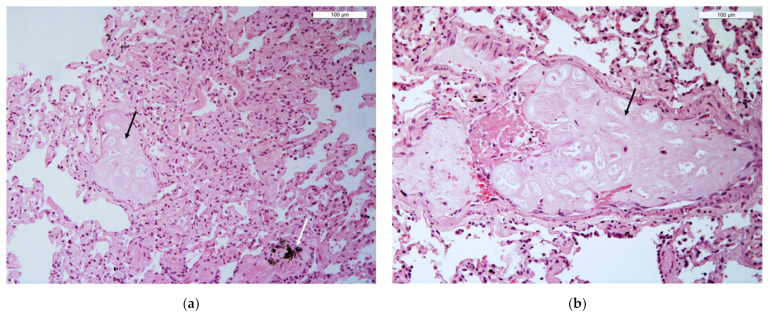
A comparison of histological slides of TBLC (**a**) and a VATS lung biopsy (**b**) with parasitic lesions within the pulmonary vessels (black arrows) caused by *Angiostrongylus vasorum*. These images show how the quality of both TBLC and SLB samples is comparable. In particular, the alveolar spaces appear suitable, as they are not collapsed or emphysematous. Additionally, there is no atelectasis induced by the method, the interstitium appears preserved, and there is no artificial extravasation of blood. The white arrow in (**a**) indicates accumulations of anthracosis (black deposits). (Ob. 20×). H.E stain.

**Figure 5 animals-12-01388-f005:**
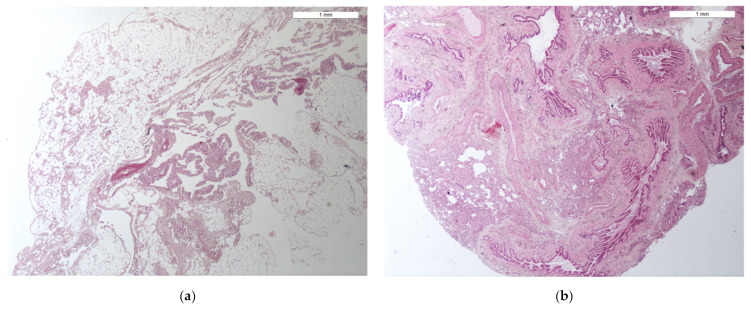
Non-diagnostic tissues found in some specimens. (**a**) VATS biopsy. Pleura (visceral and/or parietal) and thoracic wall were present in 80–100% in a large series of biopsies (40%). (**b**) One cryobiopsy consisting exclusively or mostly of several small/middle bronchus and large vessels. (Ob. 2.5×) H.E stain.

**Table 1 animals-12-01388-t001:** TBLC and VATS lung biopsies comparison. Macroscopic two-dimensional measurements, morphometric area of the histological sections, diagnostic adequacy, and related artifacts present were recorded for each sample.

CASE	TBLC LL × ll (mm)	VATS LL × ll (mm)	TBLC Morphometric Area (µm^2^)	VATS Morphometric Area (µm^2^)	TBLC Diagnostic/ Non-Diagnostic	VATS Diagnostic/ Non-Diagnostic
CASE 1 8-year-old, female, 25 kg, German Shepherd dog	5 × 4	11 × 7	9,396,718.853	408,094,846.000	yes	yes
	6 × 4	10 × 6	10,324,954.857	188,784,016.243	yes	yes
	6 × 4.5	6 × 3	12,140,963.542	14,741,529.047	yes	yes
	7 × 4	6 × 4	14,106,518.510	13,670,544.785	yes	yes
	6 × 4	7 × 5	9,986,131.659	26,115,367.832	yes	yes
	6 × 5	10 × 4	10,704,181.184	24,115,367.852	yes	yes
	7 × 4	12 × 9	7,428,219.660	29,827,114.240	yes	not (Pleura, fat 100%)
	5 × 4	10 × 7	8,475,004.000	38,789,118.787	yes	not (pleura)
	5 × 4	10 × 6	9,095,582.381	21,318,396.660	yes	Not (pleura, fat 90%)
	8 × 5	7 × 6	15,025,663.235	16,778,496.575	yes	yes
CASE 2 4-year-old, male, 20 kg, Breton dog	6 × 4	6 × 5	1,983,314.112	15,245,819.671	yes	yes
	4 × 3	11 × 5	8,943,111.418	22,510,133.198	yes	yes
	6 × 5	6 × 5	14,874,303.049	9,141,015.205	yes	Not (pleura 100%)
	7.5 × 4.5	6 × 5	22,810,981.388	19,253,547.092	yes	Not (pleura, fat 100%)
	10 × 6	9 × 7	36,447,651.581	24,082,825.732	Not (bronchus 80%)	Not (pleura 90%)
CASE 3 10-year-old, male, 22 kg mixed breed	6 ×4	4 × 3	13,795,552.710	7,606,530.042	/	/
	7 × 4	4 × 4	14,982,551.040	9,761,885.312	/	/
	8 × 4	11 × 5	7,421,963.532	22,196,350.990	/	/
	6 × 5	15 × 10	5,016,010.512	38,943,399.500	/	/
	7 × 4	6 × 2	176,640,219.500	7,413,050.998	/	/
	7 × 4	6 × 4	14,789,211.350	11,842,863.000	/	/

**Table 2 animals-12-01388-t002:** A morphometric and morpho-histological analysis of TBLC and VATS lung biopsies. The average area of macroscopic and morphometric measurements and percentages of non-diagnostic samples.

Lung Biopsy	Mean Area (mm^2^) Macroscopic (LL × ll)	Mean Area (mm^2^) Morphometric	Non-Diagnostic
VATS	48 (12–108)	46.20	40% (6/15)
TBLC	28 (12–60)	20.20	6.6% (1/15)

## Data Availability

Not applicable.

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
