# Peer review of "Comparison between Image-Guided Transbronchial Cryobiopsies and Thoracoscopic Lung Biopsies in Canine Cadaver: A Pilot Study"

_animals, 2022, doi:10.3390/ani12111388_

Round 1

Reviewer 1 Report

General comments

The article is generally well written and present a novel technique of lung biopsy in dogs.

Statistical analysis would be interesting to add value to the article.

Specific comments

Abstract:

Line 29: why do you add “s” to abbreviation “TBLCs” and not to “SLB”? Later in the text, you use only “TBLC”. Same for “ILD” (line 31), later in the text, you use “ILDs”. Please, be consistent throughout the text. Probably easier without “s” at the end throughout the text.

Line 32: What do mean by “image-guided” : precise which imaging techniques are uesd

Introduction :

Line 68 : Is doppler echocardiography the only technique to assess pulmonary hypertension in patient with ILD? Recent study states that CT angiography could also be used. As your study describes a CT technique, it should be added.   

Could you, please, introduce video-assisted thoracoscopic surgery in the different type of lung biopsies?

Could you, please,  introduce the different types of imaging guidance you use in your study?

Materials and Methods :

Line 103 and 116: How were acquired the pre- and post- CT scan examinations of the 2 fresh cadavers: in sternal recumbency? with positive pressure/inflation?

Line 108: Which parameters were used for CT-fluoroscopic guidance (slice thickness, window width, window level, artefact reduction?...)

Line 112: How were chosen the sampling sites? How many samples were taken?

Line 118: Please, use only one abbreviation throughout the materials and methods and results sections for VATS lung biopsy: don’t interchange VATS and SLB.

Were SLB and TBLC obtained in the same room or cadavers were changed in position? Were SLB and TBLC obtained on the same side of the lung and same area of the lung (peripheral vs peribronchovascular?)? Were SBL obtained under positive pressure? Did you aim to assess the same area with the 2 techniques?

Line 122: “peripheral lung”: was it for SLB? If yes, make a paragraph for SLB. It would be clearer if you better differentiate the 2 types of biopsy with 2 different paragraphs or to better clarify what is true for the 2 techniques.

Line 131: Were statistical analysis performed to assess the significance of the difference between the 2 techniques concerning these different parameters? 42 samples should permit statistical analysis.

Results :

What was the size/weight of the 3 cadavers?

Line 154: Which type of radiation protection devices were used (double side lead apron?)? Did you cover hands with lead gloves? Did you register hand exposure?

Line 186: Why 8 samples were non-diagnostic/inadequate? Could you better explain what was consider inadequate?

Line 191: how many TBLC consisted solely or mainly of bronchial wall or medium-sized vessels (% of TBLC)? When compare to VATS, was TBLC better to assess parietal pleura?

Did you assess the presence of adverse reactions locally after the procedures by opening the chest? If not, it should be added in the limitations

Discussion :

Some abbreviations used in introduction are not used in this section or are added once again in parentheses after the written-out form.

Line 234 : Could you, please, add a comment concerning radiation safety in this paragraph? What is recommended in the literature concerning real-time CT fluoroscopy and radiation safety/rules? As you describe a new technique, it is important to draw attention on this aspect.

Another point: is apnea induced in human medicine to do TBLC? Are breath-hold or positive pressure techniques recommended for pre- CT examination? What about the selection of your slices/targets for the CT fluoroscopy: could it be in a different location (slice number) on CT examination and during CT fluoroscopy?    

Could you, please, compare CT fluoroscopy and regular fluoroscopy for guidance of TBLC? What about in human medicine?

Line 249: Is bleeding time or other tests recommended before doing this procedure in human medicine? In veterinary medicine?

Line 270: could you, please, better explain your thought?

No mention in discussion of the size of the veterinary patients? Could TBLC also available for toy breed dogs? For cats? What about children in human medicine?  

References :

Some citations used abbreviated journal name, some citations used non abbreviated journal name

Figure 1 :

Could you, please, better explain the difference between images C and D and add the windowing of these images.

Figure 2 :

Could you, please, add a SLB picture to better compare the 2 techniques?

Figure 4 : could you, please, use the same magnification for the 2 techniques? Could you add the coloration used? Could you, please, add arrows, legend… to show parasites, vessels, alveolar space…

Reviewer 2 Report

1- INTRODUCTION

The drafting of the intruduction is based almost exclusively on a study published by Reinero, C. “Interstitial Lung Diseases in Dogs and Cats Part I: The Idiopathic Interstitial Pneumonias.” The Veterinary Journal 329 2019, 243, 48–54, regarding the proposed reclassification of interstitial lung diseases. The considerations justifying the purpose of the project are not fully evident in the sources cited. The citation n° 1 ( A Case of Atypical Diffuse Feline Fibrotic Lung Disease)  does not discuss the importance of biopsies and cannot be useful in supporting the experimental design. The citation n° 4( Idiopathic Pulmonary Fibrosis in West Highland White Terriers) does not report on the importance of ante mortem biopsy diagnosis and cannot support the experimental hypothesis.It is therefore essential in the introduction to verify what the evidence-based indications of the literature are related to the need to use lung biopsies for interstitial pathologies in which other methods are not able to define the diagnosis.

2 MATERIAL AND METHODS

  1. The methods proposed must be identical in all subjects, the different control methods performed to define the deepness of the probe could constitute a bias on the characteristics of the sample obtained.
  2. It is necessary to consider for each group of biopsies and in each subject the number of significant and not significant samples, especially as regards the presence of only bronchial tissue as an indicator of incorrect estimation of the sample depth.
  3. RESULTS

     A. The presence of haemorrhage cannot be considered a reliable data           since the method is cadaveric and the appearance of blood at the sampling site may not be a consequence of vascular ropture.

    B. The presence of pneumothorax judged to be mild is an unacceptable parameter in a cadaveric test since in non-ventilated subjects and with biopsies on non-elastic lung tissue its presence cannot indicate complication.

4 DISCUSSION

Discussion includes CT fluoroscopy which is not presented in the materials and methods, but only cited in the results.

5 CONCLUSIONS

The statement “Interstitial lung diseases in veterinary medicine are underdiagnosed and characterized by an important diagnostic gap due to the high risks associated with the procedures currently available” is not supported by the cited literature.

Round 2

Reviewer 1 Report

Dear authors,

Thank you for amendments.